# Ultra-Processed Food Impairs Bone Quality, Increases Marrow Adiposity and Alters Gut Microbiome in Mice

**DOI:** 10.3390/foods10123107

**Published:** 2021-12-15

**Authors:** Tamara Travinsky-Shmul, Olga Beresh, Janna Zaretsky, Shelley Griess-Fishheimer, Reut Rozner, Rotem Kalev-Altman, Sveta Penn, Ron Shahar, Efrat Monsonego-Ornan

**Affiliations:** 1School of Nutrition Science, Institute of Biochemistry, Food Science and Nutrition, The Robert H. Smith Faculty of Agriculture, Food, and Environment, The Hebrew University, Rehovot 7610001, Israel; tamara.shmul@gmail.com (T.T.-S.); olga.andryushche@mail.huji.ac.il (O.B.); janna444@gmail.com (J.Z.); shelley.griess@mail.huji.ac.il (S.G.-F.); reut.rozner@mail.huji.ac.il (R.R.); s94735net@yahoo.com (S.P.); 2Koret School of Veterinary Medicine, The Robert H. Smith Faculty of Agriculture, Food, and Environment, The Hebrew University, Rehovot 7610001, Israel; rotem.kalev@mail.huji.ac.il (R.K.-A.); ron.shahar1@mail.huji.ac.il (R.S.)

**Keywords:** bone marrow adiposity, growth plate, bone development, osteoblast, osteoclast, chondrocyte, microbiome, ultra processed food

## Abstract

Ultra processed foods (UPF) consumption is becoming dominant in the global food system, to the point of being the most recent cause of malnutrition. Health outcomes of this diet include obesity and metabolic syndrome; however, its effect on skeletal development has yet to be examined. This project studied the influence of UPF diet on the development and quality of the post-natal skeleton. Young female mice were fed with regular chow diet, UPF diet, UPF diet supplemented with calcium or with multivitamin and mineral complex. Mice fed UPF diet presented unfavorable morphological parameters, evaluated by micro-CT, alongside inferior mechanical performance of the femora, evaluated by three-point bending tests. Growth-plate histology evaluation suggested a modification of the growth pattern. Accumulation of adipose tissue within the bone marrow was significantly higher in the group fed UPF diet. Finally, microbiome 16SrRNA sequencing was used to explore the connection between diets, gut microbial community and skeletal development. Together, we show that consumption of UPF diet during the postnatal developmental period alters the microbiome and has negative outcomes on bone parameters and bone marrow adiposity. Micronutrients improved these phenotypes only partially. Thus, consuming a wholesome diet that contributes to a healthy microbiota is of a great significance in order to achieve healthy skeletal development.

## 1. Introduction

Food processing is the alteration of foods from the state in which they are harvested or raised to make them appropriate or available for consumption. For some foods, minimal processing is needed in order to make them more accessible, convenient or safe [1]. However, with the advance of mechanization and the acceleration in development of food science techniques, the main purpose and nature of food processing has been shifted from preserving whole foods towards production of a vast range of palatable products made from inexpensive ingredients [1]. Thus, food supplies recently are being dominated by heavily processed ready-to-eat products [2]. Essentially 75% of all world food sales are of processed foods [3]. In the United States (US) alone, ultra processed food (UPF) comprise 57.9% of energy intake and contribute to 89.7% of the energy intake from added sugars [4]. Concurrent with these trends, children’s UPF intake has increased markedly, with over 50% of the children in Europe and North America consuming UPF daily [5]. With the exception of provided energy, UPF have a very small nutritional value. They do not store adequate levels of fibers, vitamins or minerals, especially required for proper and healthy development of children [6].

Consequently, excessive consumption of UPF may contribute to malnutrition by inducing deficiencies of micronutrients alongside excess of energy, thus leading to unfavorable clinical outcomes [7]. A link between UPF consumption and bone quality has been observed in humans. It was established that children living next to community stores, as opposed to UPF outlets, have significantly higher bone mineral density (BMD) by dual-energy X-ray absorptiometry (DEXA) at 4 years of age [8]. Our group has recently published evidence suggesting the direct unfavorable effect of the UPF on the growing skeleton in a pre-clinical model of rats [9].

Classic malnutrition –undernutrition due to scarcity of food or malabsorption—was broadly recognized to cause longitudinal growth retardation and short stature (stunting) [10]. However, another manifestation of malnutrition—overnutrition, and the effect of excessive consumption of UPF were only recently recognized as a possible candidate of skeletal impairment [9]. Nutritional components are fundamentally important during this crucial period of growth [11]. Thus, to maximize growth and peak bone mass (PBM), nutrition must be optimized and maintained throughout this stage of rapid development. The accrual of bone mass is a major determinant of PBM and it directly impacts the risk of osteoporotic fractures during advanced age [12].

The impact of UPF on skeletal parameters may be in part mediated by modifications of the gut microbial community. The microbiota composition can change rapidly upon dietary changes [13]. Alterations in these microbial communities have been associated with a variety of conditions such as inflammatory bowel diseases (IBD) [14], obesity [15], diabetes [16], cardiovascular disease [17] and bone homeostasis [18]. In the study reported here, the effect of UPF diet was assessed by bone morphology and microarchitecture analysis, mechanical integrity evaluation and 16SrRNA sequencing of the cecum microbiome to explore the connection between diet, microbiota and skeletal development.

## 2. Materials and Methods

### 2.1. Experimental Design

64 female C57BL/6 mice after weaning (3 weeks old) were purchased from Harlan Laboratories (Rehovot, Israel). Mice were housed in cages of 4 animals per cage under standard environmental conditions with a 12 h light:12 h dark cycle and *ad libitum* access to food and drink. After 4 days of acclimation, the mice were randomly divided into experimental groups. All procedures were approved by the Hebrew University Animal Care Committee # AG-23342-01. Throughout the experiments, body weight, body length from the tip of the nose to the end of the tail, food consumption was measured twice a week (Figure A1, Appendix A), and daily food intake in kilocalories was calculated for each mouse per day. At two time points, after 3 and 6 weeks of the experiment (6 and 9 weeks of age, respectively), animals were anesthetized with isoflurane, blood samples were collected, and they were sacrificed. Their internal organs and bones (femur, tibia) were harvested. The femora were manually cleaned of soft tissue and stored at −20 °C until micro-computed tomography (µCT) scanning and mechanical testing. Tibiae were fixed immediately after sacrifice (for histological studies).

### 2.2. Ultra-Processed Food Diet

The ultra-processed food diet was purchased at a popular franchisee processed food restaurant and included a roll of bread, hamburger, tomatoes, lettuce, ketchup (excluding onion and pickles) and chips. The whole meal was homogenized, shaped as dumplings and frozen at −20°C. The diet was analyzed by an independent analytical laboratory service Aminolab, Rehovot. The results are presented in the Appendix A section Table A2. The four diets supplied during the experiment were as follows:

(1) Control diet (*n* = 16): 18.6% protein, 6.2% fat and 44.2% carbohydrate. The control diet was supplied by Harlan Laboratories Teklad Global 18% Protein Rodent Diet;

(2) Ultra-processed food diet (*n* = 16): 23% protein, 38% fat and 39% carbohydrate;

(3) Ultra-processed food diet with calcium supplement (*n* = 16): 23% protein, 38% fat and 39% carbohydrate; supplemented with calcium phosphate (MP Biomedicals, LCC) according to Harlan’s calcium content of the control chow diet;

(4) Ultra-processed food diet with multivitamin and mineral supplement (*n* = 16): 23% protein, 38% fat and 39% carbohydrate; supplemented with multi vitamin and mineral (AIN-93-VX Vitamin Mix and AIN-76 Mineral Mixture), while the calcium content was the guideline for supplementation and was equal to control and the 4th group diets.

### 2.3. Serum Hormonal Analysis

Serum samples were sent to American Medical Laboratories (Herzliya, Israel). In order to obtain the hormonal profiling of insulin, leptin, osteoprotegrin and sclerostin at both time points of the experiment after 3 and 6 weeks (at 6 and 9 weeks of age, respectively). Blood was collected by cardiac puncture upon euthanization. Multiplex kit technology was employed using LUMINEX Mouse Bone Kit (Cat. No. MBNMAG-41K).

### 2.4. Intraperitoneal Glucose Tolerance Test (IPGTT)

To determine the effect the UPF diet on glucose tolerance an IPGTT was performed using glucose in amount of 2 mg per gram of weight at 6 weeks of age. The mice were fasted for 6 h prior to the injection and glucose values at baseline (0 min), 15, 30, 60, 90 and 120 min post injection were obtained. Mice were tail-bled, and glucose values were measured using a Free-Style Optium glucometer and strips. Area under the curve (AUC) values were calculated in GraphPad software.

### 2.5. Histological Analysis

Growth plate (GP) of the proximal tibiae was examined by histological techniques at 6 and 9 weeks of age. Tibiae were fixed overnight in 4% paraformaldehyde (PFA, Rehovot, Sigma, USA) at 4 °C followed by 2 weeks of decalcification in Ethylenediaminetetraacetic acid (EDTA). Next, tibiae were dehydrated and processed as described in previous publications [19,20]. Transverse tissue sections of 5μm were prepared with Leica microtome (Agentec, Rehovot, Israel).

Safranin-O staining was performed as described elsewhere [21,22]. The sections for all histological analysis were dried and mounted with DPX mounting for histology. For imaging Eclipse E400 Nikon light microscope with DP71 camera was used, controlled by Cell A software (Olympus, Israel). Following Safranin-O staining, the thickness of each GP was measured in Cell A (Olympus) software with a measuring tool feature at 11 selected locations throughout the GP. The zones selected for measuring of the GP included the resting, proliferating and the hypertrophic zone.

### 2.6. Morphological Analysis

Left femora was scanned in a SkyScan 1174 X-ray computed microtomograph scanning device at 6 and 9 weeks of age. During scanning, femurs were moisturized with sealine and wrapped in plastic wrap to retain moisture and prevent excessive drying. After the scan bones were returned to dampened KIMTECH^®^ Science lab wipes and wrapped with plastic wrap until mechanical testing. Images were obtained with an X-ray tube voltage of 50 kV and current of 800 μA. The distal femora were scanned using a 0.25-mm aluminum filter with a 4500 ms exposure time and at high spatial resolution (13.8 μm). For each specimen, a series of 900 projection images were obtained with a rotation step of 0.4° by averaging 2 frames for a total 180° rotation. For analyses of the diaphyseal cortical region 200 slices, corresponding to 6.113–7.618 mm were chosen. Global grayscale threshold levels for the cortical region were between 110 and 255 were selected. For the trabecular region a total of 103 slices, corresponding to 2.168–2.944 mm were selected, and adaptive grayscale threshold levels between 66 and 255 were used. For all the regions, a crossectional reference slice (slice zero) was set as the first slice on which the growth plate was observed [9,23]. The data provided for bone components included data for trabecular and for cortical bone. For trabecular bone, data included: bone volume fraction (BV/TV), trabecular number (Tb.N), trabecular thickness (Tb.Th), and trabecular separation (Tb.Sp). For the cortical bone, data included: total cross-sectional area (Tt.Ar), cortical bone area (Ct.Ar), cortical area fraction (Ct.Ar/Tt.Ar), average ortical thickness (Ct.Th) and bone mineral density (BMD).

### 2.7. Mechanical Testing

The mechanical properties whole femora from each group were determined by three-point bending tests of left femora, after morphological analysis using a custom-made micromechanical testing device [24] at 6 and 9 weeks of age. The resulting load-displacement curves were used to calculate bone stiffness, yield load, and maximal load [25].

### 2.8. Osmium Staining of Marrow Adipose Tissue

Osmium staining was performed on the femur at 6 and 9 weeks of age. Right femurs were fixed in 4% neutral buffered formalin at 4 °C and decalcified in EDTA. The proximal femur was cut to allow better penetration of osmium. Marrow fat was stained with a 1:1 solution of 2% aqueous osmium tetroxide (OsO_4_) and 5% potassium dichromate (K_2_Cr_2_O_7_). Region of interest (ROI) selected for the evaluation was the metaphyseal area of the femur, above and below the GP. Fat volume in the chosen medullary compartment was quantified using high-resolution microcomputed tomography (Skyscan 1174 X-ray computed Microtomography scanner device). CT slices were evaluated beginning 1 mm below the growth plate and extending 350 slides up covering the area of 2.26 mm in total. Stained marrow fat was segmented from the adjacent marrow using a fixed threshold. Adipose volume (mm^3^), marrow volume (mm^3^), and adipose/marrow volume fraction (%) were measured [26,27].

### 2.9. Microbiome Analysis

At the day of the sacrifice, caecum samples were collected at 9 weeks of age and stored at −80 °C until the day of the extration. 0.25 gr of defrosted caecum content were extracted using QIAamp PowerFecal kit (Qiagen, Hilden, Germany) according to the manufacturer’s instructions. 16S rRNA Sequencing were performed by the University of Illinois at Chicago Core for Research Informatics (UICCRI), using single-end reads with a length of 272bp with coverage of ~38,500 per sample. Sequencing trimming was done by cutadapt, Amplicon Sequence Variants (ASVs) were inferred using DADA2 and taxonomy was classified against the SILVA 132_16S database. A biom files for each experiment was received.

Bioinformatics analysis of genus level was conducted by the Bioinformatics Unit, Weizmann Institute of Science (Rehovot, Israel). The threshold for significant differential expression (DE) was set at: adjusted *p*-values for multiple testing by Benjamini-Hochberg procedure ≤ 0.05, |log2FoldChange| ≥ 1 and max count ≥ 30 [28]. Principal coordinate analysis (PCoA) was calculated by using bray-curtis distances matrix. PCoA is an ordination analysis that simplifies high dimensional data into 2D plot. The axes of the plot are ranked in order of importance: differences along the first principal component axis (PC1) are more important than differences along the second principal component axis (PC2). In order to evaluate the diversity in each of the tested caecum samples Alpha diversity was calculated using the Shannon index. Differential abundance analysis was carried out using Deseq2 on the counts produced by Washington’s lab [29].

### 2.10. Statistical Analysis

All data are expressed as the mean ± SD (standard deviation). The significance of differences between groups was determined using JMP 12.0 Statistical Discovery Software (SAS Institute 2000) by the Tukey–Kramer HSD. Differences were considered significant at *p*  ≤  0.05. Bartlett’s test of homogeneity of variances was used to check the homogeneity of data.

## 3. Results

### 3.1. Mice Fed UPF Consumed More Energy, Gained Weight and Presented Metabolic Syndrome Characteristics

To study the link between UPF consumption and postnatal skeletal development, we conducted a six-week long in-vivo trial in young (3-week old to 9-week old) C57BL/6 female mice. Animals were divided to four groups: control diet, UPF diet, UPF diet supplemented with calcium (UPF + Ca) and UPF diet supplemented with multivitamin and mineral complex (UPF + MV). The time-frame of the experiment extends from weaning to puberty, and therefore represents the growth period before sexual maturation and growth plate (GP) closure in humans.

Energy intake of the mice were measured during the experiment. Mice tend to monitor their energetic intake [30]. However, the amount of food as well as total energy consumed by mice of the UPF groups during the 6 weeks of the experiment was substantially higher than that of the control group (Figure A2).

Expectedly, high caloric intake was followed by weight gain. At the end of the experiment mice fed UPF presented higher body weight compared to the control. As seen in Figure 1A, after 6 weeks on the UPF diet (at 6 and 9 weeks of age, respectively), UPF group mice gained more weight compared to control group. Total weight gain per mouse from the UPF group throughout the experiment was 7.1 gr and 4.3 gr in the control mice (Figure 1B), with multivitamin and mineral group resembling the control.

After 6 weeks of consuming UPF (at 9 weeks of age), the UPF diet groups had significantly elevated leptin values compared to control (Figure 1C). These findings highlight the accumulating effect of the UPF diet and correlate with weight gain of the mice as leptin is found to be elevated in obese mice [31]. Possible metabolic syndrome in the UPF group, was confirmed by elevated glucose levels (Figure 1D) and intra-peritoneal glucose tolerance test (IPGTT) (Figure 1E). The area under the curve (AUC) of the control group was substantially lower than of the UPF group or the UPF+Ca group (Figure 1F). Interestingly, UPF+MV group presented the lowest AUC values following the control group (Figure 1E,F), suggesting a favorable effect for the micronutrients. Nevertheless, there was no statistical difference. The challenged reaction of mice to glucose was accompanied by impaired insulin levels (Figure 1G), which is positively concurrent with basal plasma glucose values (Figure 1E).

Taken together, these results may indicate metabolic shift to insulin resistance and weight gain, making this model relevant for the research of skeletal outcomes accompanying this pathological state. 

### 3.2. Growth Pattern of Mice Fed UPF Was Altered

The longitudinal growth of mice assessed by body length, nose to tail, was measured at the beginning (3 weeks old), the middle (6 weeks old) and at the end (9 weeks old) of the experiment. Length of mice in all the UPF groups compared to the control group was significantly shorter at 6 weeks of age, with the gap closing at 9 weeks old and resembling the control in all the UPF groups. (Figure 2A). Since the growth plate (GP) is the engine of longitudinal growth, we performed Safranin-O staining of tibial GP at both time-points of the experiment (3 and 6 weeks of the experiment, at 6 and 9 weeks of age, respectively). No visual differences were found between the groups, GP seemed to have a regular structure with appropriate cell morphology (Figure 2B). Nevertheless, measurement of the GP thickness revealed differences in the GP width between the groups. At the end of the experiment none of the UPF groups succeeded to achieve the GP progress toward closure as observed in the control group (Figure 2C). This observation may indicate that growth pattern of mice was indeed altered by consumption of UPF.

### 3.3. Mice Fed UPF Presented Unfavorable Morphological Parameters of Their Long Bones

MicroCT scanning was used to evaluate the bone morphological parameters. The trabecular parameters of the femora were compromised in the mice consuming UPF diet. At 9 weeks of age the UPF group tended to present the lowest bone volume of total volume (BV/TV%) ratio as compared to all the other groups (Figure 3A,D). The number of trabeculae within the bone was substantially lower and the separation between them was the highest in the UPF group at the age of 9 weeks old (Figure 3B,C). These findings are consistent with each other since the bone volume within the trabecular space is driven from the number and thickness of the bone trabeculae and the spaces between them. Trabecular thickness did not differ between the groups (data not shown), thus the difference in bone volume originated from the number and the separation of the trabeculae (Figure 3B–D).

Cortical mean total area (T.Ar) of bones harvested from UPF fed mice was larger at the end of the experiment (significant for calcium and multivitamin groups compared to control), suggesting that the diameter of the bones was greater, thus the bones were bigger in size (Figure 4A,G). Nevertheless, cortical thickness tended to be lower in the UPF group compared to control at both time points of the experiment (Figure 4B). On the contrary, the medullary area in the UPF mice tended to be larger compared to control, explaining the bigger bone mean total area (Figure 4C). BMD was not affected by the various diets and ranged between 1.05 and 1.12 gr/mm^3^.

In accordance with the morphological parameters, serum osteoprotegrin (OPG) levels in the UPF groups were significantly lower than in control mice at 6 weeks of age (Figure 4D). OPG that usually inhibits osteoclast formation and activity was lower in the UPF groups, leaving them free to resorb bone. This may have resulted in the observed larger medullary area [32].

To evaluate the primary mechanical role of the skeleton three-point-bending tests of the femora were performed [24]. As could be expected by the reduction of cortical thickness in UPF group, the mechanical performance was inferior to the control group. At 6 weeks of age (only after 3 weeks of the UPF diet), yield point of the UPF group was substantially lower with the same trend observed at 9 weeks of age after 3 additional weeks (Figure 4E). Likewise, the slope of the linear part of the force-deformation curve, which represents the stiffness of the whole bone, was lower in the UPF group, especially compared to the UPF+MV at the end of the experiment (Figure 4F). Taken together, these results display the deterioration in the mechanical performance of the bone. Calcium supplementation tended to recover the compromised mechanical behavior of the UPF group, while supplementation of the UPF diet with multivitamin and mineral complex presented substantial benefit for the mechanical performance of the bone.

This data taken together suggest a negative effect of the UPF diet on the growing skeleton in both cortical and trabecular compartments of the femur, and as a result inferior mechanical performance.

### 3.4. Increased Marrow Adiposity Was Observed in Femurs of Mice Fed UPF

Adipose compartment of the bone marrow was assessed with technique that couples osmium tetroxide histochemical staining of lipid residing in the bone marrow with micro-CT for three-dimensional quantification of marrow adipose tissue (MAT) [26]. We observed an increase in MAT within the UPF groups at the end of the experiment as compared to the control group (Figure 5A,B). The multivitamin and mineral supplementation group presented a notable recovery in adipose accumulation within the bone marrow, whilst the calcium supplemented group was not as efficacious in this reclamation (Figure 5A,B). Notably, after only 3 weeks of consuming UPF diet, mice had substantially elevated sclerostin levels compared to the control group (Figure 5C). Elevated sclerostin levels in sync with enlarged MAT area is a phenomenon observed during ageing [33]. As sclerostin is proposed as a driver of marrow adiposity [34], these data are in accordance with our observation.

### 3.5. Mice Fed UPF Sustained Alterations of Their Gut Microbial Community

In order to assess the specific changes in the mice intestinal microbiota, 16S rRNA sequencing was preformed from cecum samples of 9 weeks old mice. In order to evaluate the diversity of each sample, Alpha diversity was calculated by Shannon index. Compared to the control, both supplemented UPF groups had a significantly higher Alpha diversity (Figure 6A). Interestingly, Alpha diversity between control and UPF group did not differ at all. However, significant Alpha diversity was observed in the UPF+MV group as compared to the UPF and to the control. Principal coordinate analysis (PCoA) was used to evaluate the ecological distance between samples and produce a graphical configuration in a low-dimension. Beta diversity PCoA plot based on Bray-Curtis distances matrix, was calculated on the genus level. As visible on the plot, all the groups are very different from each other (Figure 6B). The control group is completely separated from all the other UPF groups indicating its distinction. Beta diversity of the UPF group is seemingly lower, as presented by small distances in the UPF samples that mostly cluster together. Calcium or multivitamin supplemented groups resemble control group appearance, dispersed over a designated area of the plot indicating more abundant Beta diversity.

In an effort to identify the specific bacteria possibly mediating the effect of UPF consumption on the skeletal system, we focused on the significant bacterial changes comparing UPF to the control group on the phylum, family and genus levels. We also examined the effect of supplemented UPF diets on the abundance of these bacteria. Genera abundance was first assessed using a heat map (Figure 6C). For instance, *Bifidobacterium* and *Parasutterella* were more abundant in the UPF group compared to control. Moreover, the expression of *Akkermansia* was higher in both of the supplemented UPF groups compared to control. Additionally, Rikenellaceae_RC9_gut_group and Acetatifactor were less expressed in the UPF group compared to control. All in all, the microbial organisms in the intestines of mice fed UPF were very dissimilar to the control group. On a family level, Clostridiales_vadinBB60_group was less abundant in all the UPF groups as compared to control. In addition, Bacteroidetes phylum expression was lower in all the UPF groups compared to the control. Firmicutes phylum frequency was lower in the UPF group compared to the control. Furthermore, Verrucomicrobia phylum was expressed more in both of the supplemented groups (UPF+MV and UPF+Ca). Finally, the frequency of Actinobacteria was increased in UPF and UPF+MV groups compared to control. All data considered, we observed that consumption of UPF diet has indeed altered the gut microbiome of mice.

## 4. Discussion

This work shows the impact of UPF consumption on skeletal development of mice throughout the period of the first 21–72 days of their lives. Mice after weaning were chosen to imitate the pre-pubertal period of growth approximately relevant to ages 2–13 in children. Bones of mice fed UPF displayed inferior mechanical properties, a deteriorated micro-structure and increased MAT as compared to mice fed a regular chow diet.

Adiposity of the bone marrow is consistent with existing literature, as MAT accumulation is recognized to be increased by high fat diet (HFD) [35]. Our experimental UPF is high fat as it contains 38% fat of total Kcal, compared to 6.2% of the control diet. Typically, the term HFD is used in literature to describe diets containing 60% of fat and higher [36]. Unlike the conventional HFD fat content, we believe the UPF diet we used is more representative of actual consumption in terms of macronutrient distribution. Thus, even with lower fat content than the typical experimental HFD, UPF diet modified adipose metabolism. MAT accumulation has been associated with osteoclastogenesis, through increased inflammatory cytokine production, osteoclast formation and differentiation. These effects ultimately lead to higher bone resorption [37]. This finding is supported by the enlarged medullary area which was observed in all the groups fed UPF, suggesting that the equilibrium of bone remodeling has been shifted in favor of bone resorption, occurring on the endosteal bone surface surrounding the marrow [32].

After only 3 weeks of UPF consumption mice had substantially elevated sclerostin serum levels, compared to the control group. Sclerostin is a glycoprotein secreted by osteocytes, a negative regulator of bone deposition, through inhibition of osteoblast function [38]. This may indicate that osteoblasts were over-inhibited in the UPF groups, preventing them from properly constructing bone tissue. This finding corresponds with the morphometric bone parameters, as deterioration was observed in the cortical and the trabecular compartments. Multivitamin or calcium supplementation of the UPF diet did not seem to have an effect on the sclerostin serum levels.

Sclerostin levels in diabetic patients are reported to be elevated independently of age and inversely related to bone turnover markers [39]. Hence, it is reasonable to assume that the increase in sclerostin levels may have been caused by the altered glucose response triggered by consumption of UPF diet. Circulating sclerostin in humans is associated with metabolic syndrome [40], as was also observed in our experiment. Moreover, elevated sclerostin levels of the UPF groups were in accordance with enhanced MAT accumulation within the bone marrow. Recent investigation highlights the phenomenon of lipid accumulation in the marrow in conjunction with an elevation of serum sclerostin [41]. Mesenchymal bone marrow cells give rise to osteoblasts, adipocytes or chondrocytes. Sclerostin has shown to inhibit osteogenic commitment of mesenchymal cells, thus possibly favoring adiopogenesis [42]. The exact mechanism behind this shift is yet to be comprehended.

Correspondingly, mechanical properties of the femur in our experiment revealed deteriorated performance. This outcome was consistent with higher serum ALP levels of the UPF group (Table A1) as high serum levels of ALP were reported to be associated with bone pathologies [43] and fractures [44].

The microbial community in the gut of mice was altered by consumption of UPF diet. Interestingly, Alpha diversity between control and UPF groups did not differ at all. However, significant change in Alpha diversity was observed in the UPF+MV group as compared to the UPF and to the control, suggesting a positive effect of multivitamin supplementation on the diversity of the microbiota.

Bacteroidetes phylum frequency was lower in all the experimental groups as reported upon consumption of HFD [45] and in the state of obesity [46]. In addition, bone fracture incidence is significantly higher in low Bacteroides subjects as reported previously [47]. Firmicutes phylum frequency was lower in the UPF group, similarly to patients with osteoporosis [48]. Interestingly, current research supports decline of this phylum in anorexia nervosa (AN) [49], which also exhibit increase in MAT in comparison to healthy weight controls [50], coherent with our model. Thus, it is likely to assume that Firmicutes may play a role in mediating bone adiposity. Verrucomicrobia phylum was expressed in higher levels in both of the supplemented groups (UPF+MV and UPF+Ca) and lower levels in the UPF group that presented obesogenic and insulin resistant symptoms. Similarly, this phylum along with Proteobacteria is less abundant among type 2 diabetic patients [51] and individuals with obesity [52].

On a family level, Erysipelotrichaceae expression was increased in all UPF groups compared to control. As previously reported [53], an increase of Erysipelotrichaceae was observed in mice on high-fat or western diet and it is correlated to inflammation as there is strong evidence supporting its association with host lipid metabolism. Clostridiales_ vadinBB60_group along with some other groups were reported to be enriched after prolonged UVB exposure in subjects with vitamin D insufficiency, making it a mediator of serum vitamin D increase [54]. In our experiment its expression was downregulated in all of the UPF groups compared to the control. As vitamin D has a strong impact on bone health, it is possible to assume that downregulation of this group may have had an adverse effect on bone structure and function as observed in our experiment.

On the genus level, *Bifidobacterium* and *Parasutterella* were more abundant in the UPF group compared to control. *Parasutterella* expression was found to be associated with advanced IBD [55] and hypertriglyceridemia-related acute necrotizing pancreatitis [56]. Both conditions are of inflammation etiology, with 50% of IBD patients experiencing at least one extra-intestinal manifestation among them is the involvement of reduced bone density [57].

*Akkermansia* expression was higher in both of the supplemented UPF groups compared to control. This strain is associated with a healthier metabolic status and better clinical outcomes after caloric restriction in overweight/obese adults [58]. *Akkermansia* abundance is inversely proportional to bodyweight of animals and humans [59]. *Akkermansia* has been characterized as a beneficial player in body metabolism and has great prospects for treatments of metabolic disorders associated with obesity, through the production short chain fatty acids (SCFAs) in the gut, as these can have beneficial effects on glucose and lipid homeostasis [59]. In term of bone homeostasis, in a model of calvarial infection, *Akkermansia* decreased inflammatory cell infiltration and periodontal bone destruction [60]. It is reasonable to assume that supplementation of the UPF diet with calcium or multivitamin helped in upregulation of *Akkermansia* and recuperated the metabolic state of mice in these groups. They indeed weighted less than the UPF group and had a better glycemic response to IPGTT.

*Rikenellaceae_RC9_gut_group* was reported decreased at hyperglycemic related acute necrotizing pancreatitis in rats [56]. Accordingly, it was less expressed in the UPF group of our experiment, which presented compromised glycemic homeostasis as obtained with IPGTT and serum glucose and insulin measurements. *Acetatifactor* was also decreased in the UPF group compared to control. *Acetatifactor* is acetate and butyrate producing bacteria associated with type 1 diabetes [61]. *Acetatifactor* can increase glucagon like protein 1 (GLP-1) secretion directly and indirectly [62], which plays a vital role in post-meal insulin secretion and appetite suppression and thereby affecting glucose metabolism [62]. It has been reported that GLP-1 can enhance BMD and improve bone quality by promotion of bone formation and inhibition of bone resorption [63]. As *Acetatifactor* abundance was lower in the UPF group, but not in the supplemented UPF groups, it is possible that supplementation assisted in maintaining of a healthier gut, glucose homeostasis and bone morphology in these groups.

We speculate that changes in the gut microflora caused by UPF diet may interfere with changes in the skeletal system. More research with use of SPF or gnotobiotic animals is needed to fully understand these mechanisms.

## 5. Conclusions

In light of the results presented above, we propose that UPF may indeed have an undesirable effect on skeletal development through altering the microbiome and enhancing adiposity of the marrow. This, in turn, may lead to lower PBM and inferior skeletal performance in later life. As we all are exposed to diets which include such foods, it is essential to explore the long-lasting effects they may have on the skeletal network on a clinical level. Thus, high-scale human research is needed to fully understand how UPF influence the gut microflora and skeletal system development axis.

The years 2016–2025 are designated by the UN as the Decade of Nutrition, in support of the UN Sustainable Development Goals, co-led by the World Health Organization and the Food and Agriculture Organization. Thus, we find our project relevant and in sync with the UN vision and goals. Malnutrition comes in multiple forms, it requires considering food as a whole nutritious commodity, intended not only to provide energy but to assure proper development and growth mediated by many factors one of which is most probably the microbiome.

## Figures and Tables

**Figure 1 foods-10-03107-f001:**
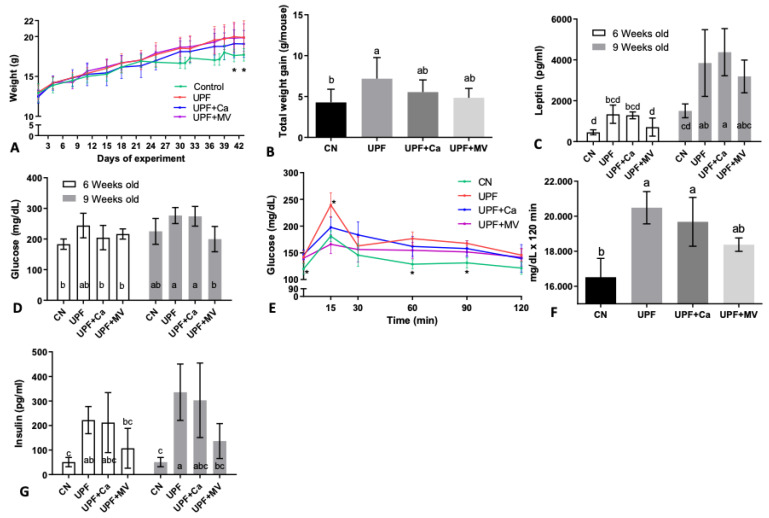
UPF effect on metabolic condition of mice at 6 and 9 weeks of age. (**A**) Weight throughout the experiment (gr) * refers control to UPF group only (**B**) Total weight gain throughout the experiment (gr/mouse) (**C**) Serum leptin (pg/mL) (**D**) Serum glucose (mg/dL) (**E**) IPGTT Intra peritoneal glucose tolerance test (mg/dL/min) at age of 6 weeks (**F**) Average area under the curve of IPGTT (intraperitoneal glucose tolerance test) (mg/dL × 120 min) (**G**) Serum insulin (pg/mL). Values are expressed as mean ± SD, *n* = 8. Different letters denote *p* < 0.05 between groups, * *p* < 0.05 compared to control. CN–control, UPF–ultra processed food, UPF+Ca-ultra processed food supplemented with calcium, UPF+MV-ultra processed food supplemented with multivitamin and mineral complex.

**Figure 2 foods-10-03107-f002:**
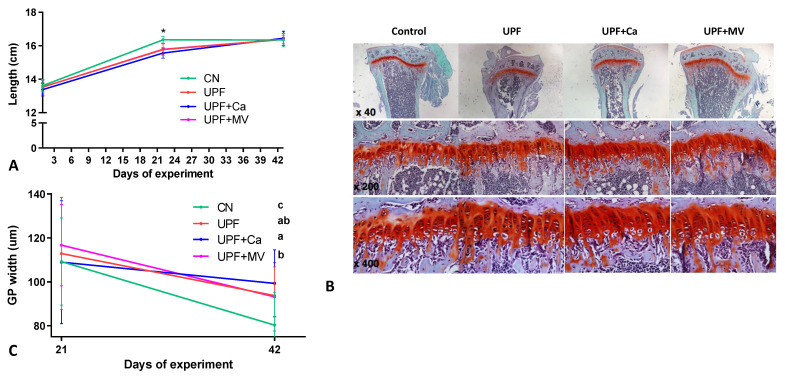
UPF effect on longitudinal growth of mice at 6 and 9 weeks of age. (**A**) Total length from nose to tail (cm) (**B**) Tibiae from the different groups were dissected, processed and stained with safranin-O at 9 weeks of age (end of the experiment) (**C**) Growth plate thickness (μm) of each GP was measured at 11 selected locations throughout the GP. The zones selected for measuring of the GP included the resting, proliferating and the hypertrophic zone. Values are expressed as mean ± SD, *n* = 8. Different letters denote *p* < 0.05 between groups, * *p* < 0.05 compared to control. CN–control, UPF–ultra processed food, UPF+Ca-ultra processed food supplemented with calcium, UPF+MV-ultra processed food supplemented with multivitamin and mineral complex.

**Figure 3 foods-10-03107-f003:**
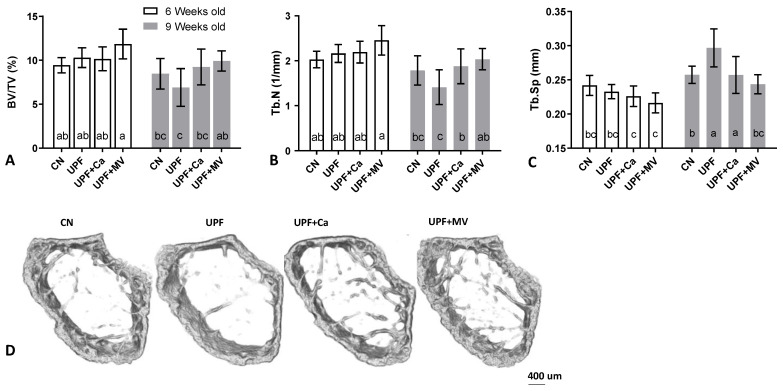
UPF effect on trabecular compartment of mice at 6 and 9 weeks of age. (**A**) Trabecular bone volume (%) (**B**) Trabecular number (1/mm) (**C**) Trabeclular separation (mm) (**D**) 3D reconstruction of trabecular compartment at 9 weeks of age. Values are expressed as mean ± SD, *n* = 8. Different letters denote *p* < 0.05 between groups. CN–control, UPF–ultra processed food, UPF+Ca-ultra processed food supplemented with calcium, UPF+MV-ultra processed food supplemented with multivitamin and mineral complex.

**Figure 4 foods-10-03107-f004:**
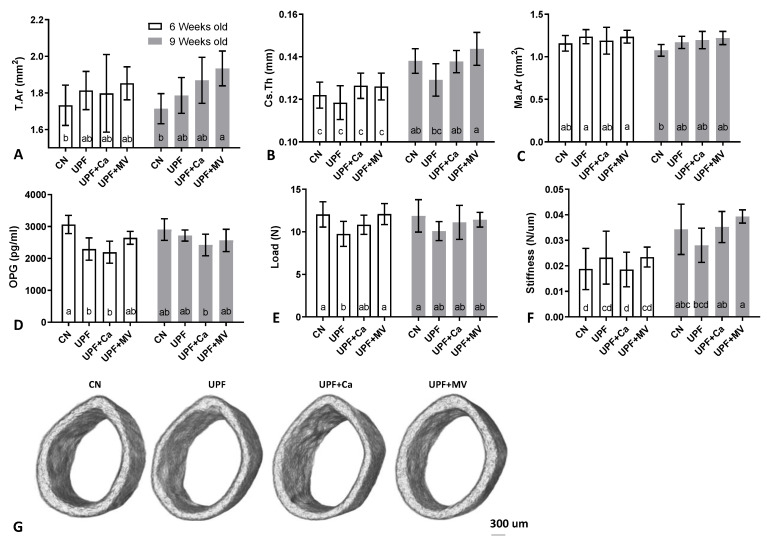
UPF effect on mechanical integrity and cortical compartment of mice at 6 and 9 weeks of age. (**A**) Mean total area (mm^2^). (**B**) Crossectional thickness (mm) (**C**) Medullary area (mm^2^) (**D**) Serum osteoprotegrin (pg/mL) (**E**) Yield load (N) (**F**) Linearity slope (n/um) (**G**) 3D reconstruction of cortical shell at 9 weeks of age. Values are expressed as mean ± SD, *n* = 8. Different letters denote *p* < 0.05 between groups. CN–control, UPF–ultra processed food, UPF+Ca-ultra processed food supplemented with calcium, UPF+MV-ultra processed food supplemented with multivitamin and mineral complex.

**Figure 5 foods-10-03107-f005:**
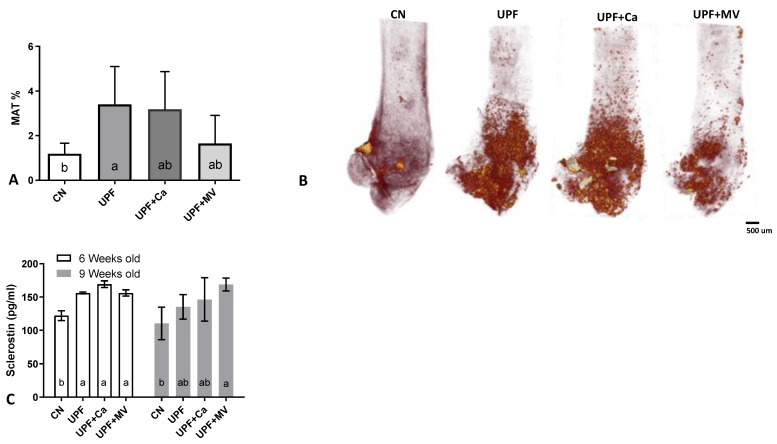
UPF effect on marrow adipose tissue of 9 weeks old mice. (**A**) Whole bone marrow adiposity (%) (**B**) 3D reconstruction of marrow adipose tissue after osmium tetroxide staining (**C**) Serum sclerostin (pg/mL). Values are expressed as mean ± SD, *n* = 8. Different letters denote *p* < 0.05 between groups. CN–control, UPF–ultra processed food, UPF+Ca-ultra processed food supplemented with calcium, UPF+MV-ultra processed food supplemented with multivitamin and mineral complex.

**Figure 6 foods-10-03107-f006:**
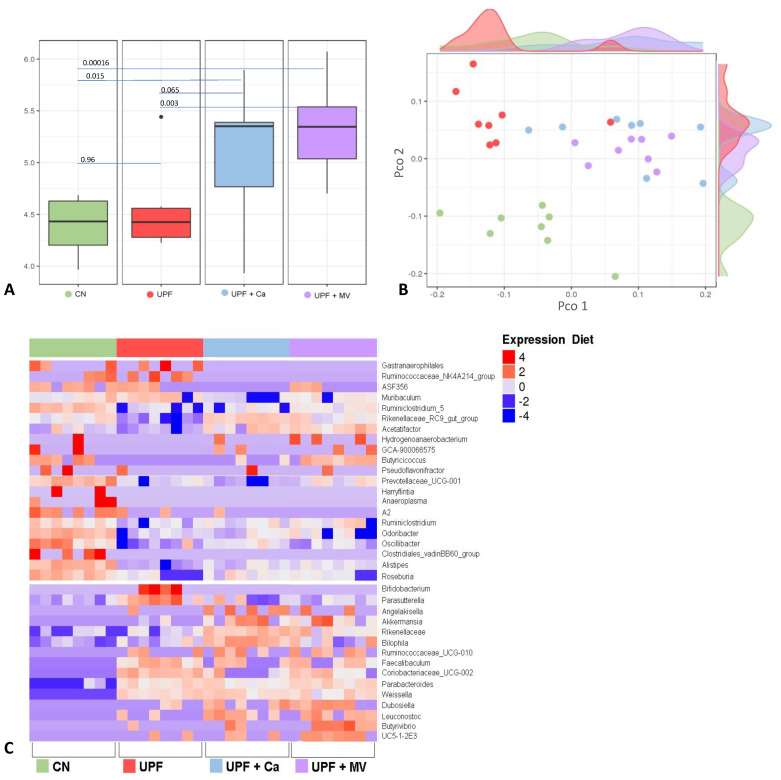
UPF effect on microbiome changes of the gut of mice at 9 weeks of age. (**A**) UPF diet effect on the bacterial diversity within samples by using Shannon index. *p*-value was calculated using Pairwise comparisons wilcox.test. (**B**) The UPF diet effect on the distance between samples based on Bray–Curtis distances matrix. (**C**) Changes in the bacterial profile, results were analyzed using Deseq2 and assessed using heatmap. CN–control, UPF–ultra processed food, UPF+Ca-ultra processed food supplemented with calcium, UPF+MV-ultra processed food supplemented with multivitamin and mineral complex.

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
