# Peer review of "Ultra-Processed Food Impairs Bone Quality, Increases Marrow Adiposity and Alters Gut Microbiome in Mice"

_foods, 2021, doi:10.3390/foods10123107_

Round 1

Reviewer 1 Report

The present study aimed to investigate the influence of highly processed food on skeletal system and intestinal microbiota in a  mouse model.

To obtain the results authors constructed four different diets which then were administered to the mice for 6 consecutive weeks. Basic serum biochemistry and serum hormonal measurement were performed in two time points, i.e. 3 and 6 weeks after the beginning of the experiment. In this time, mice were 6 or 9 week old. In addition, authors introduced standard histology of tibia bones and morphological analysis of femoral bones by using computed tomography. Finally, sequencing was applied in order to assess gut microflora of the control animals and those that have been subjected to experimental diet.

As a consequence authors achieved novel and interesting results, in which they stated that ‘ultra processed food’ is able to induce changes in bone morphological parameters, unfavorably modifies growth pattern of the bones and alters the gut microbiome which could be associated with the skeletal remodeling.

Highly processed food has been identified as a factor of many diseases in the 21st century. Consumption of these types of products is linked with obesity, diabetes or cancers. This not only causes measurable damage to human health, but also places a significant burden on the public health services involved in the treatment of diseases caused by inappropriate and unbalanced diet. In this context, the aim of the present work is fully reasonable.

However, despite of these issues (mentioned above), this reviewer (TR) would like to draw the authors’ attention to some important points. All these issues should be treated as major concerns in this article.

Introduction:

Authors wrote: ‘..the main purpose and nature of food processing has been shifted from preserving whole foods towards production of a vast range of palatable products made from inexpensive ingredients’.  This is not entirely true and correct statement. In fact, many other aspects in food industry play a role, when new foods are constructed. I recommend remodeling or removing this sentence.

Materials and methods:

  1. In general: Essential information has been omitted. For instance: How long did this experiment last? When the samples were taken (time points). Even if these data were provided in the result section (mainly on graphs), this should also be presented in m&m section. Please rewrite this section again and add the subsection entitled ‘experimental design. You should add there all necessary information about the animals and duration of the experiment.
  2. There is not enough information about the ‘UPF’ diet. If TR understood it correctly, authors used some kind of hamburger combined with chips (fast food), and then homogenized that. In addition to information about the level of protein, fat and carbohydrates, there should also be information about Mg, Ca, Na and K content. Similarly, please provide exact information about Ca and vitamin supplementation (the amounts).
  3. Please indicate in the subsection entitled ‘serum hormonal analysis”, the hormones which have been measured and the time points of the measurement. Please describe how the blood (from the mice) was collected
  4. When IPGTT test was performed? After 3 or 6 weeks, counting form the beginning of the experiment? These detailed data should be added in materials and methods as well. All information about the time points of particular analyses should be added in the appropriate subsections of the materials and methods.
  5. The abbreviation should be expanded when it appears for the first time in the text. Please check all of them in your manuscript.
  6. Statistical analysis: because authors used the parametric statistics, please provide information about the statistical tests that have been introduced to check the data normality and homogeneity of the variances.

Results:

  1. In general, in the description of the results, please clearly indicate the statistically significant differences of particular parameters between particular groups. For instance: You should not write ‘the UPF diet groups had elevated leptin values’. Was it statistically significant? If yes, please indicate in relation to which group.
  2. Graphs: sometimes you give the day of the experiment and other times day or week of the mouse’s life. Please standardize it.
  3. statistical differences (or lack of such differences) were not marked on some charts (especially bar graphs), i.e. 1G- 6 weeks old, 3A, 3B, 3C 4A-6 weeks old, 4B-6 weeks old, 4C- six week old, 4D-six week old, 4F-six week old
  4. Subsection 3.1. Authors wrote: ‘Interestingly, UPF + MV group presented the lowest AUC values following the control group (Figure 1E,F), suggesting a favorable effect for the micronutrients’. Yes, but there were no statistical differences between this group and the remaining groups.
  5. Subsection 3.2. Authors wrote: ‘Length of mice in all the UPF groups compared to the control group was significantly shorter at 6 weeks of age, with the gap closing at 9 weeks old at the end of the experiment (Figure 2A)’ It is difficult to understand and follow your way of thinking. Please write if it is statistically different or not as compared to the control group in particular time points.
  6. Graph 2C: shouldn’t be 21 instead of 22 and 42 instead 44 on x-axis?
  7. Subsection 3.3. Authors wrote: ‘Within the 6 weeks end point, the UPF group tended to present the lowest BV/TV% ratio as compared to all the other groups (Figure 3A,D)’ However, there are no information about the statitics on bar graph (3A). There is only the letter c in UPF group and nothing more (9 weeks old).
  8. Authors wrote: ‘Nevertheless, cortical thickness tended to be lower in the UPF group compared to control at both time points of the experiment (Figure 4 B)’ It does not correspond to the graph (for 9 week old animals: there are lack of statistical differences between UNF and control; nothing is known about 6 week old animals)
  9. Authors wrote: ‘On the contrary, the medullary area in the UPF mice tended to be larger compared to control, explaining the bigger bone mean total area (Figure 4 C).’ Again, If I understood it corretly, it is not true. According to your graph (4C), there are no statistcial differences between UPF group (ab) and control (a) in the case of 9 week old animals. There are no information about 6 week mice.
  10. Authors wrote: ‘In accordance with the morphological parameters, serum osteoprotegrin (OPG) levels in the UPF groups were lower than in control mice (Figure 4 D)’ In which group? 6 week animals or 9 week animals???
  11. Figure 5 A – How old were the animals for this analysis? Please provide appropriate information in the figure legend.
  12. Table A3. Please provide data as mean and SD. Please indicate (for each value) the result from statistical analysis.

Discussion and conclusion:

  1. Authors speculate that changes in the gut microflora caused by diet may interfere with changes in the skeletal system. It is possible, however, studies which use appropriate SPF or gnotobiotic animals are needed to fully understand these mechanisms.
  2. Please discuss how can we translate these results into humans? Please try to indicate their limitations. In fact, in case of human medicine and dietetics high-scale research are needed to try to fully understand how highly processed food influence on correlation between gut microflora and skeletal system development.

Summing up, this manuscript was prepared carelessly, as if in a rush. Results should be described with greater care. There are many mistakes that prevent the correct perception of this manuscript and its understanding as well. The charts do not have or have incomplete information from the statistical analysis. The description of the results does not correspond to the charts, or, the charts do not include the information described in the text. It is necessary to eliminate all these issues in order to provide high quality scientific paper.

Author Response

We would like to thank this reviewer for the meticulous reading of our paper, and the helpful suggestions. We have addressed all the comments, and believe the manuscript has significantly improved and more explanatory now.

The modifications that have been made are attached here:

Reviewer 2 Report

I have reviewed the manuscript entitled “Ultra-processed food impairs bone quality, increases marrow adiposity and alters gut microbiome in mice” for a publication in Foods as an original article. After careful review of this manuscript, I admit that some interesting results can be observed. However, Authors didn't draft the paper well, and it require a lot of editorial work.

I have some comments to the Authors.

First at all, the Materials and Methods sections must be improved, as a lot of information is missing in this section. In this work, a completely new experiment is described, therefore the most crucial information regarding the experimental scheme must be clearly given. It cannot be presented just as “mice were housed and treated as described previously [9]”, especially as experiment described in [9] was performed on rats, while presented one is performed on mice. The information about the duration of the experiment is missing. Similarly, there is no information that mice were examined at two timepoints or that the feed intake was monitored. Were they individually housed?

The same remark applies to the diet and feeding scheme as manuscript is submitted to the journal focused of foods, to special issue about nutritional effects on growth and development.

L 86-87 unclear: How protein level if control diet could be 24% when mice were fed Global 18% protein diet?

L91 What was the level of calcium in all diets? What was the level of calcium phosphate supplementation?

In general, I think that Authors should give at least a short description of procedures in sections described in L110, L136. For example there is no information whether distal or proximal part of tibia was examined. Which femora were used for histology and which for uCT ? The information what parameters were analyzed on uCT scans is missing. Was BMD measured for cortical bone only (L271)?

Ref [19] does not give any information how these traits (stiffness, yield load, maximum load) can be calculated form load-displacement curves.  This reference is redundant here.

Statistics: A lot of information regarding the microbiome analysis is missing (Alpha diversity, Shannon index, Beta diversity PCoA plot, Bray-Curtis distances )

Please read the entire manuscript carefully. All abbreviations must be explained when first used, not only in he abstract, but also in the main text (some examples: L41 – UPF, L51 – BMD, L111 –  GP, L459 – GLP-1). Bacteria genus should be given in italics.

L53 I think that the information that in that study a rat model was used should be given.

L67-70 Rephase. it is not clear enough what was evaluated for mechanical integrity (bone) or 16s rRNA-seq (microbiome).

L94 “AIN-76 Mineral Mixture”

L99 Please list examined biomarkers here.

L104 and others – SI symbol for gram is “g” not “gr”. Please consider changing.

L112 What decalcification agent was used? EDTA, like for osmium staining?

L133 In what plane the analysis procedure was performed ?

L144 Where bones were cut off ?

L145 2% aqueous osmium tetroxide (OsO4) and 5% potassium dichromate (K₂Cr₂O₇)

L163 correct “padj” to “adjusted p-values” and “Benjamin-Hochberg” to “Benjamini-Hochberg”

Figures - try to unify the nomenclature - weeks or  days through the manuscript. as analysis were performed on two time points, I suggest the weeks.

Fig 1E,F – in figure caption there should be information about the timepoint (6 or 9 wk) of the analyses.

Fig 1A. it should be clearly marked that * refers to PUPF vs control group only

Fig 2C. No SD is given.

Fig 3A – statistical significance not clear. Why single “c” for 9-wks old mice ?

Fig 4C,D,F – Please be consistent. C and F should be without “ab” or “ab” should be added to D.

Fig 4C correct “slope” to “stiffness”

L307 osteoprotegrin

Fig 6B. Correct “PC” to “PCo”

L356 “Bray–Curtis”

L374 Suggest changing. UPF diet contains 38% of fat (as a calory source), but not 38% of total Kcal (consumed).

Some limitations of the current study should be given. For example, Authors suggests that the bone remodeling has been shifted towards bone resorption, however no markers of bone resorption were  analysed – the most useful markers of bone resorption are degradation products derived from the enzymatic hydrolysis of type 1 collagen (CTx, TNx, DPD, PD). Similarly, no Ca, P or vit D levels in blood serum were determined. No information about how the results obtained for applied pre-clinical model can be transferred to human studies is given ( When do mice reach somatic and sexual maturity? How do 6- and 9-weeks old mice relate to human growth and development?).

I hope my comments are helpful for achieving a better version of this manuscript.

Author Response

We would like to thank this reviewer for his positive attitude and important comments, we addressed all comments as detailed in the attached documents (in red) and feel that the paper was improves by these modifications.

Round 2

Reviewer 1 Report

Reviewer's questions and comments were addressed by the authors; however I see several other problems with that manuscript. 

There is still  lack of information on how the homogeneity of variance and the distribution of the data have been checked (statistical analysis section)

But the most important problem is that the Authors  superimposed new graphs over the old ones. As a result, this reviewer (TR) is unable to read the results and compare them with the description of the results in the body of the text.  Appropriate new version of the manuscript, which include only new graphs, should be prepared and added to the system.

Author Response

Thanks again for the suggestions, these the modifications we have 

There is still  lack of information on how the homogeneity of variance and the distribution of the data have been checked (statistical analysis section)

Thank you for your kind reminder, info added to M&M . Bartlett’s test of homogeneity of variances was used to check the homogeneity of data.  

But the most important problem is that the Authors  superimposed new graphs over the old ones. As a result, this reviewer (TR) is unable to read the results and compare them with the description of the results in the body of the text.  Appropriate new version of the manuscript, which include only new graphs, should be prepared and added to the system. Sorry, we fixed it

Reviewer 2 Report

The Authors have significantly improved the paper. However, some issues still need to be clarified.

The applied track changes options make it impossible to read the revised graphs. A corrected new revised version should be prepared.

BMD abbreviation stands for bone mineral density, not bone mass densitometry.

Still there is no information whether mice were kept individually

Mechanical testing. Reviewer has concerns regarding the tests performed. The whole uCT procedure took probably about 2 hours (900 images with exposure time of 4.5 s). During the whole procedure, bones were exposed to adverse environmental conditions in the apparatus, which undoubtedly led to a significant drying of the bones. This had a significant impact on the results of mechanical test, specially influencing the stiffness calculation results.

In presented study, the experiment comprises four experimental groups. Which data were analysed using t-test?

The sentence “Groups may have more than one letter to reflect the “overlap” between the sets of groups” is unnecessary, while the statement “sometimes a set of groups is associated with only a single treatment level” is not clear.

Author Response

Thanks again for the suggestions, these are the modifications we have done:

The applied track changes options make it impossible to read the revised graphs. A corrected new revised version should be prepared. Sorry, we fixed it

BMD abbreviation stands for bone mineral density, not bone mass densitometry. Corrected

Still there is no information whether mice were kept individually Info added to the M&M "experimental design, 4 mice per cage.

Mechanical testing. Reviewer has concerns regarding the tests performed. The whole uCT procedure took probably about 2 hours (900 images with exposure time of 4.5 s). During the whole procedure, bones were exposed to adverse environmental conditions in the apparatus, which undoubtedly led to a significant drying of the bones. This had a significant impact on the results of mechanical test, specially influencing the stiffness calculation results.

Thank you for your accurate remark, we added this important ifo tho m&m : During scanning, femurs were moisturized with seline and wrapped in plastic wrap to trap moisture and prevent excessive drying. After the scans, bones were returned to dampened KIMTECH®Science lab wipes and wrapped with plastic wrap until mechanical testing.

As drying is reversible, we believe if occurred, it was fixed with damp storing of the femurs before mechanical testing.

In presented study, the experiment comprises four experimental groups. Which data were analysed using t-test  Tukey–Kramer HSD test was performed in order to compare all the groups

The sentence “Groups may have more than one letter to reflect the “overlap” between the sets of groups” is unnecessary, while the statement “sometimes a set of groups is associated with only a single treatment level” is not clear. Corrected